

# Preparation and analysis of zero gases for the measurement of trace VOCs in air monitoring

Jennifer Englert[1], Anja Claude[1], Alessia Demichelis[2], Stefan Persijn[3], Annarita Baldan[3], Jianrong Li[3], Christian Plass-Duelmer[1], Katja Michl[1], Erasmus Tensing[1], Rina Wortman[3], Yousra Ghorafi[3],

Maricarmen Lecuna[4], Guido Sassi[4], Maria Paola Sassi[2], Dagmar Kubistin[1]

[1]Deutscher Wetterdienst (DWD), Hohenpeissenberg, 82383, Germany

[2]Istituto Nazionale di Ricerca Metrologica (INRIM), Torino, 10135, Italy

[3]VSL - Dutch Metrology Institute, Delft, 2629 JA, Netherlands

[4]Politecnico di Torino (POLITO), Torino, 10135, Italy

*Correspondence to:* Dagmar Kubistin (dagmar.kubistin@dwd.de)

**Abstract.** Air quality observations are performed globally to monitor the status of the atmosphere, its level of pollution and to assess mitigation strategies. Regulations of air quality monitoring programmes in various countries demand high precision measurements for harmful substances often at low trace concentrations. These requirements can only be achieved by using high quality calibration gases including high purity zero gas. For volatile organic compound (VOC) observations, zero gas is

defined being hydrocarbon free like purified air, nitrogen or helium and is essential for the characterisation of the measurement devices and procedures, for instrument operation as well as for calibrations. Two commercial and one self-built gas purifiers were tested for their removal efficiency of VOCs following a standardised procedure. The tested gas purifiers included one adsorption cartridge with an inorganic media and two types of metal catalysts. A large range of VOCs was investigated including the most abundant species typically measured at air monitoring stations. Both catalysts were able to

remove a large range of VOCs whilst the tested adsorption cartridge was not suitable to remove light compounds up to $C_4$. Memory effects occurred for the adsorption cartridge when exposed to higher concentration. This study emphasises the importance to explicitly examine a gas purifier for its intended application before applying in the field.



## 1 Introduction

Volatile organic compounds (VOCs) play an important role in atmospheric chemistry. They are key substances in the tropospheric ozone and secondary organic aerosol formation, affecting human health and climate. Main sources of VOCs are biogenic processes (e.g. plant metabolism) and anthropogenic activities (e.g. fossil fuel or industrial solvents emissions). The

variety of VOCs is enhanced by subsequent oxidation processes. Main sink process is the oxidation by the daytime cleaning agent, the hydroxyl radical (OH). Thus, the abundance of VOCs alters the self-cleaning capacity of the atmosphere and the removal of less reactive pollutants like carbon monoxide and the greenhouse gas methane.

VOC concentrations in the background atmosphere are typically at low levels of few pmol/mol up to some nmol/mol demanding for measurement techniques with very high sensitivities e.g. gas chromatography systems (GC) or state-of-the-art

proton-transfer-reaction mass spectrometers (PTR-MS). High quality zero gases are needed for determining their background signals and for performing system checks e.g. blank, memory effect and leak detections. Additionally, zero gases are essential for dynamic calibration methods since VOC calibration standards are often generated either by permeation or diffusion into a controlled zero gas stream (ISO6145-10, 2002 and ISO6145-8, 2005, Demichelis, 2016). An alternative is the dynamic dilution of a highly concentrated static standard gas mixture with a zero gas stream using mass flow controllers

(ISO6145-7, 2009). Besides, zero gases are applied for the operation of GC systems as carrier gas of GC columns and for fuel gas of flame ionisation detectors. The need for high purity zero gases is further driven by more stringent quality objectives from the WMO GAW programme (WMO GAW Report No. 171, 2007) or the ACTRIS network (Hoerger et al., 2015). These networks aim to observe the long-term trends of VOC concentrations in the background atmosphere.

A high quality zero gas is defined by containing insignificant concentrations of the target components to be measured. In

particular for VOC measurements, the hydrocarbon compounds of the zero gas have to be below the detection limit of the instruments. The highest quality commercial zero gases in gas cylinders (air, nitrogen or helium grade 5.5. or higher) are specified to contain below 10 to 100 nmol/mol total hydrocarbons. These levels far exceed the needed purity for a zero gas in atmospheric background monitoring with concentrations down to some pmol/mol. To reduce the amount fraction of VOCs, different gas purification technologies are available. Preparation has to be simple, fast and low-cost and applicable at remote

unattended stations. Furthermore, the preferred method is dependent on the VOCs present in the gas to be purified, the gas matrix and maintenance interval. Some commonly used purification technologies are:

Gas purifiers based on inorganic media

The composition of inorganic media is not disclosed by the producers (Conte et al., 2008). However from the underlying patents (e.g. EP0365490A1 and WO2013119883A1), it can be inferred that such purifiers are typically based on so-called

gettering alloys which can be either binary (such as Zr/Fe used for e.g. nitrogen purification) or ternary (like Zr/Fe/V). These getters can be operated in a wide range of temperatures including room temperature. Advantages of this technology include



the absence of electricity for operation. In addition, regeneration of several media is possible by heating the purifier to high temperatures together with flushing with clean gas.

Metal catalysts

In catalytic combustion the VOC removal relies on the complete oxidation of the VOCs to water and carbon dioxide. Two
classes of catalysts, noble metals and metal oxides have been widely used for VOC application (Liotta, 2010). The noble metal based catalysts, in spite of their higher costs, are normally preferred because of their high specific activity, resistance to deactivation and ability to be regenerated. Their catalytic performance depends on the type of VOCs, oxygen concentration, and the gas flow rate (Heck et al., 2009). The optimum conversion of the VOCs is achieved at a minimum operation temperature of about 200 to 400 °C, depending on the VOC species and the type of catalyst. For platinum or
palladium catalysts, temperatures of 400 °C are recommended by the ACTRIS "standardized operating procedures for VOCs measurements" (ACTRIS, 2014).

Photocatalytic techniques

In photocatalytic systems VOCs adsorbed on a catalyst are removed by oxidation with highly reactive hydroxyl radicals, producing ideally carbon dioxide and water as final products (Debono et al., 2013). For formation of hydroxyl radicals, a
reactor coated with a photo oxidative catalyst is irradiated with UV light in the presence of oxygen and water vapour. Titanium dioxide ($TiO_2$) is the dominant photo-catalyst because of its excellent photocatalytic oxidation ability, high photo corrosion resistance, and non-toxic properties (Huang et al., 2016). In the VOC degradation process several intermediates including carbon monoxide can be formed interfering with the complete removal of VOCs. As an example, Debono (2013) and co-workers studied the photocatalytic oxidation of decane at nmol/mol levels and observed several intermediates
including aldehydes, ketones, and alcohols.

Activated carbon

Activated carbon is processed for high porosity with an increased surface for adsorption (Van Osdell et al., 1996; Sircar et al., 1996), e.g. 1 gram of carbon can provide an internal area of around 1000 $m^2$. It can be produced from various raw materials with high carbon content including coal, wood and coconut shells. The two most common methods to activate the
carbon are chemical and steam activation. For the purification of gases it is used in the form of granulates or pellets. The adsorption of VOCs is both reversible and competitive, i.e., easily adsorbed compounds can displace weakly adsorbed components (Van Osdell et al., 1996). The efficiency typically improves with decreasing vapour pressure. Efficiency for lighter gases like methane is low.



In this study, three purifiers were selected to test their removal efficiency of a defined amount of VOCs. An adsorption cartridge with an inorganic media was selected for low-cost zero gas production without the need of electricity. In addition, the commonly used catalytic technique with an infinite lifespan has been tested for two types of catalyst.

## 2 Experimental

### 2.1 Tested purifiers and analytical methods

The tested commercial adsorption cartridge was based on inorganic media not being further specified by the manufacturer. Clean dry air (CDA) was stated to be used as the input gas with a maximum flow rate of 50 slpm. No additional heating of the purifier was required. The manufacturer claimed the removal of condensable organics below 1 pmol/mol without any further specifications of those compounds. Maximal incoming contaminant concentrations were indicated with 10 µmol/mol.

The lifetime was stated with one year at nominal flow rate with 1 µmol/mol inlet challenge of moisture. The second purifier was a commercial catalyst with 3 - 5 % palladium oxide (manufacturer SAES Pure Gas, Model PS15-GC50-CDA-2). It was specified for CDA with a maximum flow rate of 3 slpm. Its operation temperature was 350 °C. Elimination of methane and NMHCs below 1000 pmol/mol was stated by the manufacturer. Maximum inlet impurities were 2 µmol/mol total hydrocarbons. At the rated flow of 3 slpm and at rated working temperature the manufacturer stated an infinite lifespan of

the catalyst without the need of regeneration. The third purifier was a home-made metal catalyst built by the German Meteorological Service (Deutscher Wetterdienst, DWD). It consisted of a stainless steel tubing (1 inch diameter) with a length of 1 m filled with aluminium oxide pellets with 0.5 % platinum (Heraeus, Germany). The tubing was heated to 400 °C and was built in an aluminium profile box filled with perlite for thermal insulation. A stainless steel mesh (25 µm) at the end of the tubing was used for particle protection of the subsequent instruments.

The performance of the purifiers was tested by detecting residual VOC concentrations in the zero gas with gas chromatography (GC) systems. Prior to GC analysis VOC fractions were pre-concentrated either by adsorbent materials or cryogenically cooled glass beads. Subsequently, the VOCs were thermally desorbed from these traps and separated in one or more capillary columns of the GC. For detection, flame ionisation detectors or mass spectrometers were deployed. Five different GC systems were used: two for non-methane hydrocarbons (NMHCs) operated by DWD (Hoerger et al., 2015,

Plass-Duelmer et al., 2002) and the Dutch Metrology Institute (VSL), one for monoterpenes by DWD (Hoerger et al., 2015) and three for oxygenated VOCs (OVOCs) by DWD, VSL and the Istituto Nazionale di Ricerca Metrologica (INRIM). A large range of VOCs was investigated including the most abundant species typically measured at air monitoring stations as well as acetonitrile (see Table 2).



## 2.2 Experimental measurement setup and procedure

For comparability a common procedure was applied by the three labs:

[1]     Check for internal blanks of the system and in-house impurities (VOC amounts) by measuring the in-house zero gas (5 runs).

[2]     Check for VOC impurities originating from the tested purifier by measuring the in-house zero gas flowing through the tested gas purifier (5 runs).

[3]     Determine the efficiency of VOC removal by measuring a VOC mixture flowing through the tested gas purifier (5 runs).

[4]     Verify the incoming VOC concentration by measuring the same VOC mixture without gas purifier (5 runs).

After step four a repetition of steps one and two was optional for the labs but is advisable to monitor the status of the set-up.

A unified flow rate of 1 slpm was applied being within the specification of each purifier model. The two catalysts were heated and flushed with zero gas for at least two hours before starting the experiments. This was needed to reduce VOC impurities originating from the catalysts being freshly installed. The experimental set-up is shown in Figure 1.

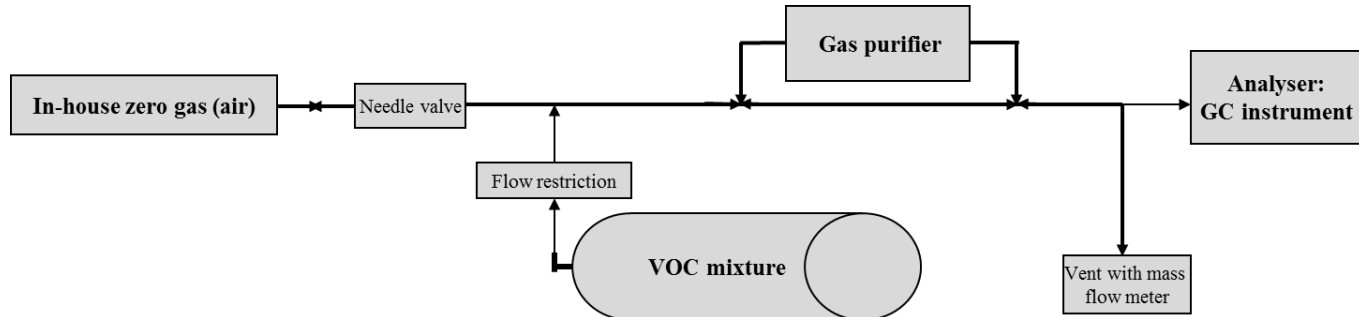

**Figure 1.** Experimental set-up for testing the purifier performance.

Test mixtures with different VOC mole fractions were produced by dynamic generation methods, e.g. dilution of high concentrated static VOC mixtures in cylinders (Figure 1) or diffusion methods (Demichelis, 2016). Following test mixtures were supplied: NMHCs at 1.2, 5 and 50 nmol/mol, monoterpenes at 1.2 nmol/mol, OVOCs from 10 to 70 nmol/mol and

acetonitrile at 10 nmol/mol. For the in-house zero gas DWD used compressed and dried (water content ~ 1000 µmol/mol) ambient air purified by a palladium catalyst. VSL and INRIM used synthetic air cylinders (grade 6.0, water content < 0.5 µmol/mol, total hydrocarbons content < 0.05 µmol/mol).





## 3 Data analysis methods for zero gas characterization

### 3.1 Quantification of VOC impurities in zero gases and handling of system internal blanks

The quantification of zero gas impurities $x_{\text{imp}}$ is given by

$$x_{\text{imp}} = (y - B)/A \tag{1}$$

where y is the measured signal of the investigated impurity [µV min] and B is the signal of the system internal blank [µV min]. The signals are defined by the integration of the detector response [µV] in the identified retention time interval [min] of the investigated VOC, i.e. the VOC peak area. A is the detector sensitivity to the investigated VOC [µV min / concentration], the so-called calibration factor.

Identification of internal blanks B, i.e. system artefacts, and discrimination of them from zero gas impurities is done by measuring different sample volumes in step 1 of the measurement procedure (Sect. 2.2). A proportional relationship of the detector response with the sampled volume is expected for impurities in the zero gas measured, whereas for GC system internal blanks the detector response is expected to be independent of the sample volume.

### 3.2 Determination of the analytical detection limit

Gas chromatography measurements at very low concentrations like in this study demand low limits of detection. The detection capability, i.e. detection limit $x_D$ is defined by IUPAC as the smallest measure that can be detected with reasonable certainty for a given analytical procedure. Other definitions of the limit of detection are reported in literature (Belter, 2014) and different approaches based on regression of gas standards are described (Shrivastava, 2011, Belter, 2014, IUPAC, 1995).

Following IUPAC, the detection limit is based on Neyman–Pearson theory of hypothesis testing (IUPAC, 1995). This definition considers the probability of false positive $\alpha$ and false negative $\beta$ detections and focuses on reducing the probabilities of making errors. The correctness of the method was proved by Voigtman (2008). $x_D$ is calculated by Currie's formula Eq. (2) (IUPAC, 1995), where a linear calibration curve is assumed and the detector signal y is described by y = B + A$x$, with the regression intercept B (blank value), the sensitivity A (calibration factor) and the analyte amount $x$.

$$x_D = \frac{2\,t_{1-\alpha,v}\sigma_0}{A}\frac{K}{I} \tag{2}$$

with

$$K = 1 + r(A,B)\frac{u(B)}{\sigma_0}\frac{t_{1-\alpha,v}u(A)}{A} \tag{3}$$

$$I = 1 - \left[t_{1-\alpha,v}\frac{u(A)}{BA}\right]^2 \tag{4}$$

$$r(A,B) = -\frac{\bar{x}}{x_q} \tag{5}$$



where $\sigma_0$ is the standard deviation of the measured system internal blank. If no internal blank is present, the standard deviation of the baseline signal is used. This value is experimentally assessed by integrating the noise of the detector over a time interval similar to the average peak width for a serial of zero gas measurements. $t_{1-\alpha,v}$ is the t-student value for $v$ degree-of-freedom (equal to n-2) and $\alpha$ level of significance (equal to 0.05 – one tail). $r(A,B)$ is the correlation coefficient

5   with $\bar{x}$ as the mean of the $n$ samples and $x_q$ as the quadratic mean. u(A) and u(B) are the uncertainties of the calibration factor and blank value, respectively. If the uncertainties of the linear calibration function parameters are negligible (IUPAC, 1995, Sect. 3.7.5.1), $\frac{K}{I} \approx 1$, equation 2 simplifies and the detection limit $x_D$ is calculated by:

$$x_D = k \cdot \sigma_0 / A \qquad (6)$$

where k = 3.29 and A is the calibration factor.

10   To improve the detection limit of a GC device several parameters can be optimised summarised in Table 1.

**Table 1.** Opportunities for detection limit $x_D$ improvement of GC measurement systems.

| Opportunities for $x_D$ improvement | Description | Actions |
|---|---|---|
| $\downarrow\sigma_0$ (when system internal blanks are detected) | Improve reproducibility of system internal blank measurements<br><br>Improve baseline noise | - realize reproducible system cleaning<br><br>- realize reproducible pre-concentration<br><br>- employ high purity carrier gas and detector gases |
| $\uparrow$A | Increase detector sensitivity<br><br>Increase the amount of compound reaching the detector and/or in the case of FID the ion production rate in the flame | - increase sampled mass (or volume) on the VOC trap<br><br>- increase the mass flow rate ratio hydrogen/air, the makeup mass flow rate (prefer $N_2$ to He), the FID temperature |
| $\downarrow t_{1-\alpha,v}$ | Increase $v$ degree-of-freedom | - increase $N$ (number of blank determinations) |
| $\downarrow u_A$ and $u_B$ | Improve regression quality | - design a suitable regression experiment in terms of number of gas standards and gas standards range (for non-linear detectors) |

In Table 2 are reported the detection limits $x_D$ for the various VOCs under test calculated with the method of IUPAC.



**Table 2.** Tested VOCs by lab with the individual detection limits in pmol/mol.

| | compound | DWD | VSL | INRIM |
|---|---|---|---|---|
| NMHCs | ethane | 3 | 20 | |
| | ethene | 7 | 21 | |
| | propane | 2 | 10 | |
| | propene | 3 | 11 | |
| | isobutane | 1 | 10 | |
| | ethyne | 10 | 15 | |
| | n-butane | 1 | 11 | |
| | trans-2-butene | 1 | 4 | |
| | 1-butene | 2 | 4 | |
| | isobutene | | 6 | |
| | cis-2-butene | 1 | 3 | |
| | isopentane | 1 | 3 | |
| | n-pentane | 1 | 8 | |
| | 1,3-butadiene | 1 | 5 | |
| | trans-2-pentene | 1 | 13 | |
| | 1-pentene | 1 | 3 | |
| | 2-methylpentane | 1 | 6 | |
| | n-hexane | 1 | 1 | |
| | isoprene | 2 | 4 | |
| | n-heptane | 1 | 4 | |
| | benzene | 2 | 3 | |
| | 2-2-4-trimethylpentane | 1 | 4 | |
| | n-octane | 1 | 5 | |
| | toluene | 6 | 4 | |
| | ethylbenzene | 5 | 7 | |
| | p-, m-, o-xylene | 5 | 6 | |
| | 1-3-5-trimethylbenzene | 6 | 10 | |
| | 1-2-4-trimethylbenzene | 2 | 16 | |
| | 1-2-3-trimethylbenzene | 2 | | |
| monoterpenes | alpha-pinene | 4 | | |
| | myrcene | 3 | | |
| | 3-carene | 2 | | |
| | cis-ocimene | 2 | | |
| | p-cymene | 2 | | |
| | limonene | 2 | | |
| | camphor | 2 | | |
| | 1,8-cineole | 5 | | |
| OVOCs | methanol | 77 | 110 | 3 |
| | acetaldehyde | 84 | 110 | |
| | ethanol | 26 | 120 | 11 |
| | acetone | 31 | 80 | 11 |
| | MEK | 2 | 180 | |
| | methacrolein | | 110 | |
| acetonitrile | - | | 6 | |





## 4 Results and discussion

To ensure comparability between the participating groups the same measurement procedure described in Sect. 2.2 has been applied. All GC chromatograms were analysed visually. Peaks of VOCs in the chromatograms were integrated by GC software and mole fraction were subsequently determined for each single measurement and average mole fractions and
standard deviations, respectively, were derived for each measurements series (Table 3).

Before assessing the purifier efficiency, blanks were determined by steps one and two of the measurement procedure (Sect. 2.2.). With step one GC internal blanks and impurities were detected by taking different sample volumes (Sect. 3.1) of the in-house zero gas. E.g. for the DWD NMHC system a small internal blank value for benzene has been observed (Fig. 2).

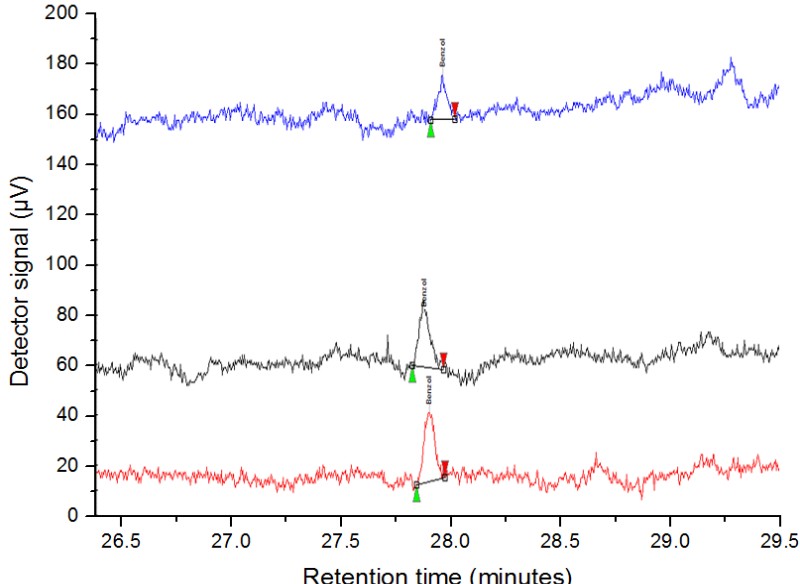

**Figure 2.** Example chromatograms for a GC internal blank of benzene (peak in flame ionisation detector signals at the retention time of benzene). Measurements of zero gas at different sample volumes: 300 ml (blue); 800 ml (black) and 1400 ml (red). The benzene peak area is independent of the sample volume.

Subsequently, VOC release of the purifier itself was checked (step 2 in Sect 2.2). E.g. the platinum catalyst showed acetaldehyde impurities scaled with the sample volume (Fig. 3). By flushing the catalysts for two hours with zero air, the
relevant impurities were below the detection limits.





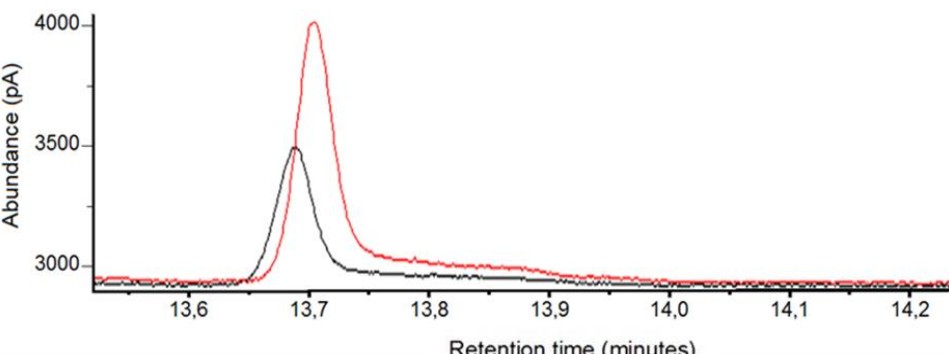

**Figure 3.** Example chromatograms for an impurity released by the platinum catalyst at an early stage of operation (abundance at the retention time of acetaldehyde). Measurements of zero gas at two sample-volumes: 680 ml (black) and 1360 ml (red). There is a proportional relationship of the detector response with the sample volume of air leaving the

purifier. In the example the acetaldehyde impurity concentration was about 300 pmol/mol.

After characterisation of the blank values, the purifier efficiencies were determined (step 3 in Sect. 2.2). In Table 3, the results (Eq. 1, Sect. 3.1) of all labs are summarised. Both tested catalysts (palladium as well as platinum) removed NMHCs and monoterpenes to concentrations below the detection limits which were generally below 10 pmol/mol (Table 2). Example chromatograms of the platinum catalyst are displayed in Fig. 4 showing an efficient performance of the purifier.

All tested OVOCs were removed to mole fractions below 100 pmol/mol. Only the lab of INRIM detected residuals of methanol and acetone above the detection limits of their system. These OVOCs are generally prone to adsorption and desorption effects on surfaces in the instruments and therefore subject to high measurement uncertainties and blank values. Consequently, detection limits are usually elevated as seen for the DWD and VSL systems (Table 2). For the INRIM system, however, rather low detection limits were indicated and no blank values were reported. Nevertheless, the fact that for both

types of purifiers and for varying input concentrations (20-70 nmol/mol) similar mole fractions for methanol and acetone (Table 3) were detected by INRIM implies the possibility that here system blanks or artefacts were observed. Unfortunately, a repetition of the blank measurements was not performed at INRIM with this set-up after this experiment and no further conclusions can be drawn.





**Table 3.** Summarized results: Mean values of 5 subsequent measurements and absolute standard deviations (± pmol/mol) of the residual amounts after purification in pmol/mol. The different amounts of VOCs the purifiers were supplied with are indicated in nmol/mol. Testing labs are specified. The detection limits of the measurement systems are indicated in the case of zero measurements (< detection limit). Residual amounts above the detection limits of the systems are marked in light grey. Values above 100 pmol/mol are marked in dark grey. N.a. = not analysed.

| Purifier: | Adsorption cartridge | | | | | | Palladium catalyst | | | | | | Platinum catalyst | |
|---|---|---|---|---|---|---|---|---|---|---|---|---|---|---|
| Supplied amount [nmol/mol]: | 1.2 | 5 | 10 | 10 | 50 | 20 to 70 | 1.2 | 5 | 10 | 10 | 50 | 20 to 70 | 1.2 | 10 |
| Testing lab: | DWD | VSL | DWD | VSL | VSL | INRIM | DWD | VSL | DWD | VSL | VSL | INRIM | DWD | DWD |
| **Residual NMHCs [pmol/mol]:** | | | | | | | | | | | | | | |
| ethane | 1063 ± 7 | 4855 ± 70 | | | 42715 ± 419 | | < 3 | < 20 | | | < 20 | | < 3 | |
| ethene | 967 ± 7 | 997 ± 231 | | | 7535 ± 505 | | < 7 | < 21 | | | < 21 | | < 7 | |
| propane | 992 ± 17 | 2764 ± 108 | | | 24513 ± 1674 | | < 2 | < 10 | | | < 10 | | < 2 | |
| propene | 703 ± 93 | < 11 | | | 26 ± 4 | | < 3 | < 11 | | | < 11 | | < 3 | |
| isobutane | 769 ± 392 | 3483 ± 1264 | | | 2258 ± 1979 | | < 1 | < 10 | | | < 10 | | < 1 | |
| ethyne | 928 ± 17 | 278 ± 73 | | | 2708 ± 538 | | < 10 | < 15 | | | < 15 | | < 10 | |
| n-butane | 675 ± 411 | 3475 ± 649 | | | 1232 ± 951 | | < 1 | < 11 | | | < 11 | | < 1 | |
| trans-2-butene | 51 ± 39 | < 4 | | | < 4 | | < 1 | < 4 | | | < 4 | | < 1 | |
| 1-butene | 195 ± 125 | < 4 | | | < 4 | | < 2 | < 4 | | | < 4 | | < 2 | |
| isobutene | n.a. | < 6 | | | < 6 | | n.a. | < 6 | | | < 6 | | n.a. | |
| cis-2-butene | < 1 | < 3 | | | < 3 | | < 1 | < 3 | | | < 3 | | < 1 | |
| isopentane | < 1 | < 3 | | | < 3 | | < 1 | < 3 | | | < 3 | | < 1 | |
| n-pentane | < 1 | < 8 | | | < 8 | | < 1 | < 8 | | | < 8 | | < 1 | |
| 1,3-butadiene | < 1 | 535 ± 183 | | | 619 ± 61 | | < 1 | < 5 | | | < 5 | | < 1 | |
| trans-2-pentene | < 1 | < 13 | | | < 13 | | < 1 | < 13 | | | < 13 | | < 1 | |
| 1-pentene | < 1 | < 3 | | | < 3 | | < 1 | < 3 | | | < 3 | | < 1 | |
| 2-methylpentane | < 1 | < 6 | | | < 6 | | < 1 | < 6 | | | < 6 | | < 1 | |
| n-hexane | < 1 | < 1 | | | < 1 | | < 1 | < 1 | | | < 1 | | < 1 | |
| isoprene | < 2 | < 4 | | | < 4 | | < 2 | < 4 | | | < 4 | | < 2 | |
| n-heptane | < 1 | < 4 | | | < 4 | | < 1 | < 4 | | | < 4 | | < 1 | |
| benzene | < 2 | < 3 | | | < 3 | | < 2 | < 3 | | | < 3 | | < 2 | |
| 2-2-4-trimethylpentane | < 1 | < 4 | | | < 4 | | < 1 | < 4 | | | < 4 | | < 1 | |
| n-octane | < 1 | < 5 | | | < 5 | | < 1 | < 5 | | | < 5 | | < 1 | |
| toluene | < 6 | < 4 | | | < 4 | | < 6 | < 4 | | | < 4 | | < 6 | |
| ethylbenzene | < 5 | < 7 | | | < 7 | | < 5 | < 7 | | | < 7 | | < 5 | |
| m-, p-, o-xylene | < 5 | < 6 | | | < 6 | | < 5 | < 6 | | | < 6 | | < 5 | |
| 1-3-5-trimethylbenzene | < 6 | < 10 | | | < 10 | | < 6 | < 10 | | | < 10 | | < 6 | |
| 1-2-4-trimethylbenzene | < 2 | < 16 | | | < 16 | | < 2 | < 16 | | | < 16 | | < 2 | |
| 1-2-3-trimethylbenzene | < 2 | n.a. | | | n.a. | | < 2 | n.a. | | | n.a. | | < 2 | |
| **Residual monoterpenes [pmol/mol]:** | | | | | | | | | | | | | | |
| alpha-pinene | < 4 | | | | | | < 4 | | | | | | < 4 | |
| myrcene | < 3 | | | | | | < 3 | | | | | | < 3 | |
| 3-carene | < 2 | | | | | | < 2 | | | | | | < 2 | |
| cis-ocimene | < 1 | | | | | | < 1 | | | | | | < 1 | |
| p-cymene | < 1 | | | | | | < 1 | | | | | | < 1 | |
| limonene | < 2 | | | | | | < 2 | | | | | | < 2 | |
| camphor | < 2 | | | | | | < 2 | | | | | | < 2 | |
| 1,8-cineole | < 5 | | | | | | < 5 | | | | | | < 5 | |
| **Residual OVOCs and acetonitrile [pmol/mol]:** | | | | | | | | | | | | | | |
| methanol | | | < 77 | < 110 | | 67 ± 11 | | | < 77 | < 110 | | 73 ± 10 | | < 77 |
| acetaldehyde | | | < 84 | < 110 | | n.a. | | | < 84 | < 110 | | n.a. | | < 84 |
| ethanol | | | < 26 | < 120 | | < 11 | | | < 26 | < 120 | | < 11 | | < 26 |
| acetone | | | < 31 | < 80 | | 57 ± 10 | | | < 31 | < 80 | | 63 ± 9 | | < 31 |
| MEK | | | < 2 | < 180 | | n.a. | | | < 2 | < 180 | | n.a. | | < 2 |
| methacrolein | | | n.a. | < 110 | | n.a. | | | n.a. | < 110 | | n.a. | | n.a. |
| acetonitrile | | | < 6 | n.a. | | n.a. | | | < 6 | n.a. | | n.a. | | < 6 |



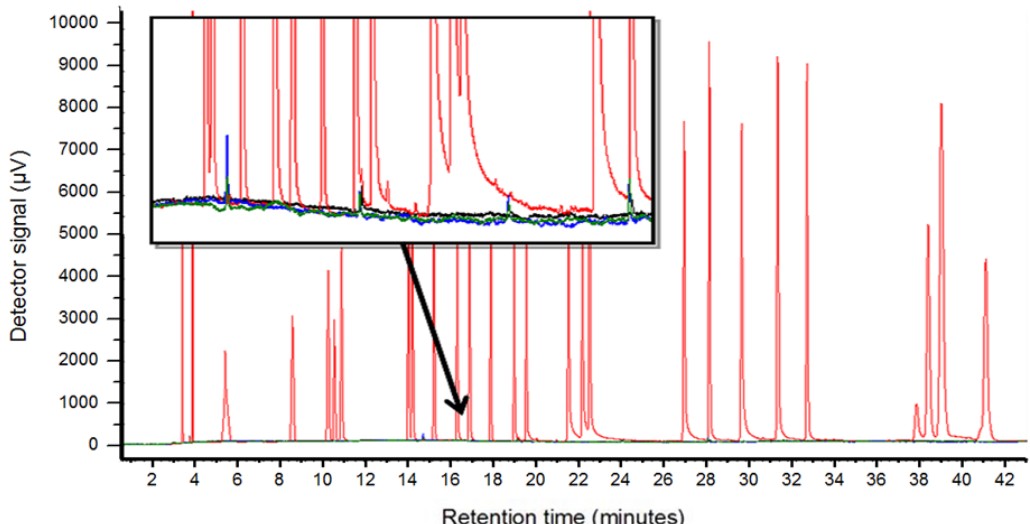

**Figure 4.** Example for the results of the catalysts, in this case the catalyst with platinum on aluminium oxide pellets (GC chromatograms): Zero gas (black), zero gas passing the catalyst (blue) confirming no relevant additional impurities are introduced by this catalyst, 1.2 nmol/mol mixture of different NMHCs (red) and the same mixture leaving the catalyst (green). Four small peaks below 10 pmol/mol are the result of system internal blanks (the same peaks are present in the measurements of zero gas). All these measurements were conducted with the same sample volume.

For the adsorption cartridge a breakthrough of light NMHCs (from $C_2$ to $C_4$) was observed by all testing labs (Table 3, Fig. 5 and 6). At a sample flow of 1 slpm ethane, ethene, propane, propene, isobutane, ethyne, n-butane, trans-2-butene, 1-butene and 1,3-butadiene were not efficiently removed as visible in an example chromatogram (green line in Fig. 5). All $C_5$ and heavier NMHCs, monoterpenes and acetonitrile were removed to values below the detection limits of the systems. For OVOCs, see the discussion of the catalyst results above.



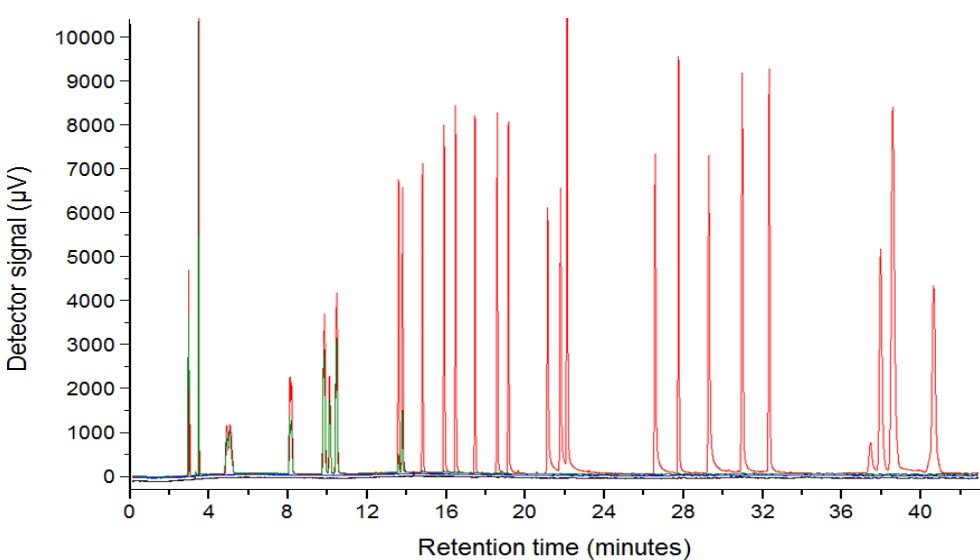

**Figure 5.** Example for the breakthrough of the light NMHCs $C_2$ to $C_4$ through the adsorption cartridge (GC chromatograms): 1.2 nmol/mol mixture of different NMHCs (red) and the same mixture leaving the catalyst (green) with a breakthrough of the light NMHCs (up to minute 14) being almost as high as the input. All these measurements were conducted with the same sample volume.

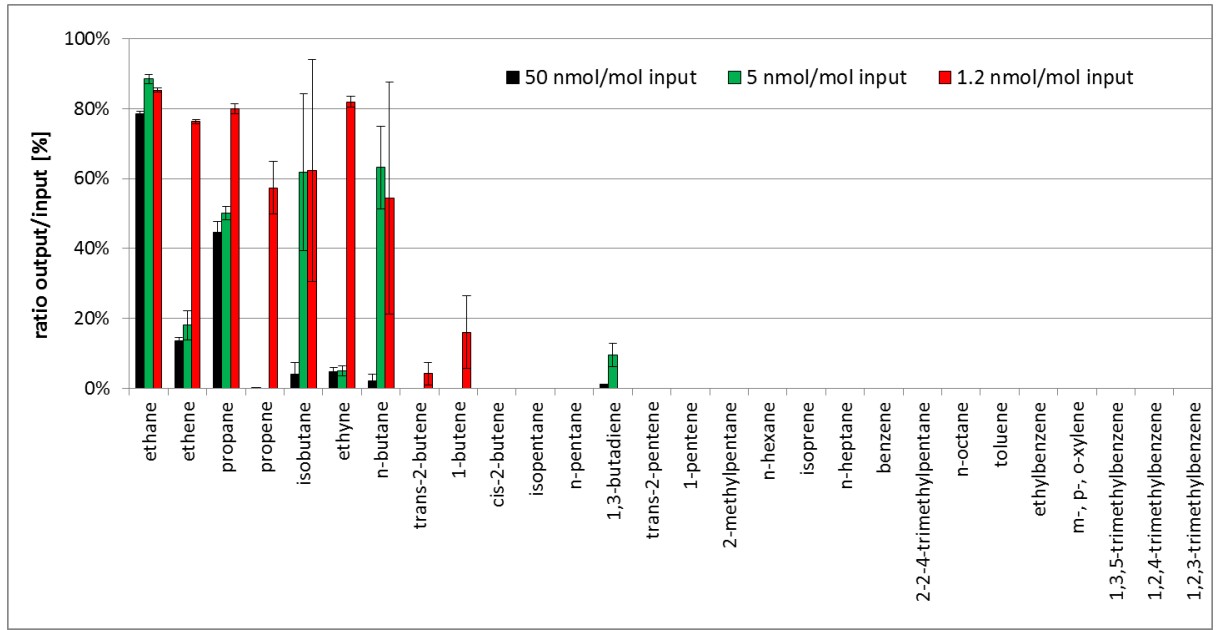

**Figure 6.** Results of the adsorption cartridge (ratios output/input) for an input of 50 nmol/mol (black), 5 nmol/mol (green) and 1.2 nmol/mol NMHCs (red). Error bars indicate the standard deviations of 5 measurements.



In Figure 6, the average output-to-input ratio for different input mole fractions and substances is shown. A ratio of 0% implies that the purifier removes the substance efficiently, whereas a ratio of 100% denotes a complete breakthrough of a substance. Except for ethane, the removal efficieny is not consistent for different input concentrations. For ethene, propane, propene, ethyne, trans-2-butene and 1-butene the 1.2 nmol/mol input was less efficiently purified compared to the higher

inputs. Several reasons are possible: First, these results were produced by two different labs which tested the same model of cartridge but not the identical cartridge. The two cartridges may show different behaviours. Furthermore, DWD responsible for the 1.2 nmol/mol experiment used a zero gas for the tests which had a much higher humidity (water content ~ 1000 µmol/mol) than the test gas from VSL which came from a commercial synthetic air cylinder (water content < 0.5 µmol/mol). The humidity level has an impact on the purifier lifetime. The manufacturer of the adsorption cartridge stated that the

humidity of the DWD zero gas would saturate this kind of cartridge almost immediately (personal communication). It should only be used with very dry air with at maximum 1 µmol/mol water content. A closer look into the individual results of the measurements series of the VOC mixture running through the adsorption cartridge reveals another effect: The breakthrough behaviour is affected by the repetition of measurements and changes with each iteration. This is reflected in high standard deviations for some substances in Table 3. In Figure 7, the results of the measurement series for 50 nmol/mol and 5

nmol/mol input are plotted for isobutane and n-butane. For the 50 nmol/mol input the output is increasing from measurement to measurement, while the reverse behaviour is observed for the 5 nmol/mol input. The latter measurements were conducted directly after the high input of 50 nmol/mol and most likely memory effects occurred. This means after supplying high VOC amounts to the cartridge, some VOCs are released by the cartridge even if the input level is reduced again.

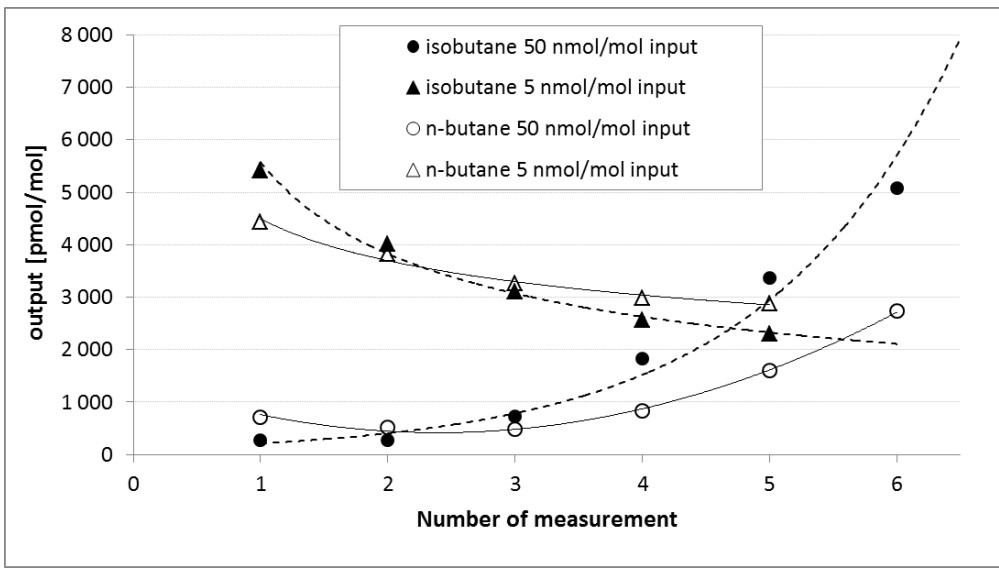

**Figure 7.** Adsorption cartridge breakthrough of isobutane and n-butane increasing with the runtime of the 50 nmol/mol mixture. A memory effect of the purifier was observed in the following measurements with 5 nmol/mol input.




# 5 Conclusions

The suitability of a gas purification system depends on the requirements for the zero gas. In a zero gas for air monitoring application, the target components to be measured have to be zero or below the detection limits of the measurement systems. It is highly important to explicitly examine a gas purifier for its intended application. Tests should be done at the given conditions, e.g. the same flow rates and the same gas matrix with special focus on given target component concentrations and humidity. For the tests, measurement systems with adequate detection limits are essential. Potential internal blanks have to be detected and well characterised. A possible blank change over time has to be monitored.

Two tested catalysts in this study were able to remove a large range of different VOCs. High mole fractions up to 50 nmol/mol were purified and residual concentrations were below the detection limits of the systems going down to less than 1 pmol/mol for NMHCs. It was shown that it is important to conduct a cleaning procedure of the catalysts as they can emit some VOCs in early stages of use. Consequently, they were flushed with zero gas and held at operation temperature for at least two hours without connecting them to the measurement instruments. Subsequently, it is necessary to check the purity of the catalysts output again. Only this way it is guaranteed to have a working set-up to characterise the purifier efficiency correctly.

The tested adsorption cartridge was not suitable to remove light NMHCs ($C_2$ to $C_4$). There was a breakthrough behaviour of these compounds which was not constant. Also, VOC memory effects were observed. To characterise these effects repetition of measurements ($> 5$) would be of an advantage. However, it removed heavier VOCs, OVOCs and monoterpenes. An advantage of the adsorption cartridge is the lack of electricity. It could be a good alternative for applications where the breakthrough of light VOCs is of no relevance. A big disadvantage is the high influence of humidity on the lifetime of this kind of purifier. The tested model in this study was only adequate for use with very dry air up to maximum 1 µmol/mol water content. With this awareness it is highly recommended to enquire the maximum applicable water content of the used gas from the manufacturer of a purifier.

Finally, zero gas is often produced by compression of ambient air which constitutes a complex matrix with residual humidity. The cleaning process to receive high purity zero gases is a challenge to any purifying system. Their long-term behaviour has to be controlled, especially for the enduring use in air quality monitoring stations.

*Acknowledgements.* The research leading to these results was performed under the European Metrology Research Programme (EMRP) project ENV56 KEY-VOCs which is jointly funded by the EMRP participating countries within EURAMET and the European Union.



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
