# Peer review of "Preparation and analysis of zero gases for the measurement of trace VOCs in air monitoring"

_Atmospheric Measurement Techniques, 2017_

## Referee Comment (RC1) · J. Rudolph (Referee) · 11 Dec 2017

The paper presents exactly what the title says, a description and characterisation of methods for the purification of air specifically designed for use in the preparation of test gases, including calibration gases, for the analysis of VOC in the atmosphere. There are numerous methods described in literature for the preparation of zero-air for exactly this and other purposes. However, as far as I am aware, these published descriptions are limited typically to brief descriptions of air-purification procedures as part of the description for various experimental set-ups and usually do not present a very detailed characterisation of the performance of the air purifier. This paper provides detailed information about the quality of zero-air for VOC analysis and efficiency of several air purification procedures and provides insight into the possibilities, limitations

and problems in the preparation of zero-air for VOV analysis. In essence the paper is a "Technical Note" of very good quality than a scientific paper. In my opinion a (substantially shortened, see below) version merits publication in AMT. My largest concern is the length of the paper. Although overall the results are clearly presented, there are substantial parts of the paper which are unnecessary and only loosely connected to the subject of the paper. I have to admit that after reading the paper I was positively impressed by the solid work that had been done, but frustrated by the fact that I had to go through 15 pages of text, figures and tables to extract some rather straightforward information that probably could be presented in a paper less than 7 or 8 pages long and a supplement. Specific suggestions: 1. Introduction: The part describing the principle of several methods for generation of hydrocarbon free air should be removed. The presented methods are not complete. For example, "pressure swing" methods and use of clean oxygen and nitrogen to prepare clean air are not mentioned, charcoal is by far not the only adsorbent used for air purification. Furthermore, the information provided is essentially textbook level and only vaguely connected to the methods tested here and no information about the performance of the different methods is provided, which greatly reduces the usefulness of this part for the reader. The explanations about the importance of clean gases in general should be removed, the paper contains no information about purifying gases other than removing non-methane VOC from air. 2. The description of steps [1] to [4] (beginning of 2.2) should be clarified. A clear description (and distinction) of "what was done" and "what was determined" at this point will allow to shorten the later (often indirect) explanations of how data were evaluated and what was found. For example, it is later explained that (as far as I understand) step one included measurements using different volumes of zero air. This needs to be explained right away (including the volumes used, after all this is the experiment chapter). 3. The procedures used to generate (and as to determine the quality, see comment 7) "in house zero air" has to be given in the experimental section. 4. Subchapters 3.1 and 3.2 should be moved to a supplement. The typical reader of such a paper will not be interested in the details of peak evaluation and DL determination and knowledge of these

details is not necessary to understand the results presented here. 5. Table 1 should be removed. Its content is only very indirectly connected to the subject of the paper and the information presented has already been discussed in numerous publications and textbooks. 6. Table 2 should be moved to a supplement. The detection limits (as far as they are relevant) are obvious from Table 3 (I assume the <... indicates a concentration below the lower detection limit, a footnote explaining this should be added). If concentrations are above the DL the DL has little relevance for the findings presented here. 7. Table 3 should be separated into 3 tables (NMHC, Terpenes, OVOCS) which will avoid the many "empty" boxes. The "saved space" should be used to present the residual VOC concentrations for the use of "in-house zero gas" as feed for the gas purifier as well as the VOC level in the "in-house" zero air without gas purifier. In my opinion, this information is of high interest for potential readers. What are the residual levels of VOC when using a "standard" combination of clean air supply and a given gas purifier. A detail for Table 3 (and some other places in the paper), the number of significant digits presented should be consistent with the accuracy of the given data. 8. The finding that catalysts have to be "cleaned" by running for some time is not new, this part should be moved to a supplement. A useful information (if a available) would be the time constant (if available) at which the different contaminants are removed from the catalysts, which would be relevant for justifying the two hour conditioning time used here. 9. Figure 6 and the detailed discussion of the results of Figure 6 should be moved to the supplement. Breakthrough as well as memory effects and dependence of the efficiency of adsorbents on humidity are nothing mew. Moreover, it seems from Figure 7 that (even when averaging) giving a value for efficiency for the adsorbent is arbitrary since the result will (least for some of the < C5) depend on the duration of exposure to a feed with a given VOC level as well as the history of exposure to feeds with different VOC levels. The low efficiency for most <C5 HC and the high variability of efficiency for most C4 HC is evident from Table 3. 10. All chromatograms should be moved to a supplement. From the chromatograms I could not gain any important insight which is not already evident from Table 3. 11. In the supplement the authors should provide

linear regression information for the results obtained for step [1] in chapter 2.2 (sampling of different volumes) for compounds where the peak areas are not below the DL. This will allow readers interested in details to distinguish between "system blanks" and signals depending on sampled volume. 12. Conclusions: The first paragraph is mostly a summary of the introduction. It also contains statements that cannot be derived from the results presented her (e.g. the importance of monitoring blanks). This paragraph should be removed.

---

## Referee Comment (RC2) · Anonymous Referee #2 · 4 Jan 2018

The paper of Englert et al. investigates purifying techniques for preparation of "zero gas" for analytical applications, e.g. analysing traces of volatile organic compounds (VOC) in air samples. Three different types of gas purifiers were tested: (1) absorption cartridge, (2) heated palladium catalyst and (3) heated platinum catalyst. The tests were conducted at three different laboratories. The air purifiers using the catalytic techniques are suitable to remove a broad range of VOC from air whereas the adsorption cartridge cannot fully clean air.

In summary, the results presented are suitable to be published in AMT but as a Technical Note after streamlining the results presented together with an Electronic Supplement. The experiments conducted are tests with already available techniques for purifying air and thus no clear innovative approach is presented which may merit a

publication as a full paper.

The "introduction" should be condensed with finally leading to the research questions. The author team should justify their selection of the used purifying techniques.

In the "experimental" part all commercial suppliers of materials should be listed. Details of the analytical systems used in the experiments should be reported in the supplement, likely in a Table.

The "data analytical details" chapter should be moved to the supplement as well as Table 1 and Table 2. A summary of the applied techniques is sufficient as part of the experimental/method section.

The "results and discussion" part should focus on the performance of the purifying systems only. Additional information should be moved to the supplement. Figures 2 and 3 can be skipped. Figures 4, 5 and 6 may also be skipped or moved to the supplement, Figure 7 should be moved to the supplement.

In the "conclusion" part of the paper please avoid duplication of the abstract and focus on aspects users of air purification systems should consider.

---

## Short Comment (SC1) · 13 Jan 2018

This paper presents the performance of several gas purifiers to produce zero gas for measuring VOCs. The availability of high purity and certified zero gases for atmospheric measurements is one of the issues that has to be tackled in order to achieve further reductions in measurement uncertainty. The information presented in this paper is interesting and it moves towards this direction; however, I miss in the Introduction section a reference to previous work in this field. There are not many papers dealing with this topic so I find it necessary to cite the few of them in order to properly contextualise the manuscript of the authors.

Some of the references I miss:

[Figure]

Miñarro et al. (2014) Zero gas reference standards. Analytical Methods 8, 15, 3014-3022.

Haerri H.P (2009) Trace gas analysis for the evaluation of zero air generators. Accreditation and Quality Assurance 14, 12, 647-654.

---

## Author Response (AR1)

**Author´s response to AMT review - first stage (Response to reviewer and short comments) Atmos. Meas. Tech. Discuss., doi:10.5194/amt-2017-412, 2017**

**"Preparation and analysis of zero gases for the measurement of trace VOCs in air monitoring"**

Jennifer Englert, Anja Claude, Alessia Demichelis, Stefan Persijn, Annarita Baldan, Jianrong Li, Christian Plass-Duelmer, Katja Michl, Erasmus Tensing, Rina Wortman, Yousra Ghorafi, Maricarmen Lecuna, Guido Sassi, Maria Paola Sassi, Dagmar Kubistin

The authors are grateful to both reviewers for their time and effort in evaluating this manuscript and for their suggestions for improvements. All points made by the reviewers are addressed on the following pages. We also thank the short comment author for his useful note. We have combined the specific responses to all comments in the following. Concerning the proposal to publish this work as a technical note, we do no oppose but would like to leave this decision to the editor.

**Referee #1**

Comment 1:

*Introduction: The part describing the principle of several methods for generation of hydrocarbon free air should be removed. The presented methods are not complete. For example, "pressure swing" methods and use of clean oxygen and nitrogen to prepare clean air are not mentioned, charcoal is by far not the only adsorbent used for air purification. Furthermore, the information provided is essentially textbook level and only vaguely connected to the methods tested here and no information about the performance of the different methods is provided, which greatly reduces the usefulness of this part for the reader. The explanations about the importance of clean gases in general should be removed, the paper contains no information about purifying gases other than removing non-methane VOC from air.*

Response:

We followed the suggestion of referee #1 and removed the part describing several methods for generating hydrocarbon free air. Instead we just listed the different principles which are applicable for atmospheric monitoring with some references for the interested reader.

p.2 lines 8ff:"Commonly used purification technologies in atmospheric monitoring include but are not limited to  gas purifiers based on inorganic media (e.g. Conte et al., 2008) or activated carbon (Van Osdell et al., 1996; Sircar et al., 1996), metal catalysts (Liotta, 2010; Heck et al., 2009) and photocatalytical techniques (Debono et al, 2013; Huang et al., 2016) ."

We did not remove the explanations about the importance of clean gases in general, as we think this is important information for understanding the need of pure zero gases and the research content of this paper. The reviewer is right that no other gas matrix than VOC-free zero air was tested for this paper. But the method described for testing this gas could be applied for other gas matrices like pure nitrogen as well as for other target analytes.

Comment 2:

*The description of steps [1] to [4] (beginning of 2.2) should be clarified. A clear description (and distinction) of "what was done" and "what was determined" at this point will allow to shorten the later (often indirect) explanations of how data were evaluated and what was found. For example, it is later explained that (as far as I understand) step one included measurements using different volumes of zero air. This needs to be explained right away (including the volumes used, after all this is the experiment chapter).*

Response:

We agree and inserted a more detailed description of what was done in step 1 to 4. We moved the explanation of the method for determining internal blanks of the analysis system from chapter 3.1 into this chapter (2.2).

p.4, lines 3ff:

**"2.2 Experimental measurement setup and procedure**
For comparability a common procedure was applied by the three labs. Sample volumes used were dependent on the requirements of the different analysers. In general, sample volumes between 400 ml to 3000 ml were applied. For all following steps a repetition of five consecutive runs was recommended:

[1]     In step 1, the in-house zero gas was measured directly by the analysis systems to quantify its VOC impurities. Additionally all analysis systems were checked for internal blanks. Identification of internal blanks, i.e. system artefacts, and discrimination of them from zero gas impurities was done by measuring different sample volumes of the in-house zero gas. A proportional relationship of the detector response with the sampled volume is expected for impurities in the in-house zero gas, whereas for GC system internal blanks the detector response is expected to be independent of the sample volume. The tested in-house zero gas was used for the following steps of the experiment (2 to 4).

[2]     In the next step the in-house zero gas from step 1 was supplied to one specific purifier to quantify the VOC impurities originating from the purifier itself.

[3]     In the third step the efficiency of VOC removal of the tested purifier was checked by supplying a VOC mixture and measuring the outcome of residual VOCs.

[4]     In the last step the incoming VOC concentration for step 3 was checked by supplying the same preparation of VOC mixture directly to the analysis system (no purifying)."

Comment 3:

*The procedures used to generate (and as to determine the quality, see comment 7) "in house zero air" has to be given in the experimental section.*

Response:

Information about in-house air is given in Section 2, p5, line 6:

"For the in-house zero gas DWD used compressed and dried (water content ~ 1000 µmol/mol) ambient air purified by a palladium catalyst. VSL and INRIM used synthetic air cylinders (grade 6.0, water content < 0.5 µmol/mol, total hydrocarbons content < 0.05 µmol/mol). "

5    The quality can be further derived from the separate in-house zero air measurements in Table 1. We added some words to the results section 3, p.6 line 6ff:

"Before assessing the purifier efficiency, in-house zero gas quality and internal blanks were determined by step one of the measurement procedure (Sect. 2.2.). The results for VSL and DWD are shown in the first two columns in Table1. In the DWD in-house zero air all substances were below the detection limit, with exception for benzene (4pmol/mol), acetaldehyde
10   (124pmol/mol) and acetone (52pmol/mol).  The observed peaks were independent of the sample volume (see supplemental Fig. S1and S2), and showed the characteristics of an internal blank and are not regarded as an impurity of the DWD in-house zero gas. For VSL, blank values were observed at a level of 20-50 pmol/mol for several alkanes (Table 1The results are consistent within the specification of the used synthetic air grade 6.0 allowing up to 50nmol/mol of hydrocarbons. This highlights the need for further purification of commercial cylinders to assure low impurities levels for high quality zero air.
15   With the INRIM system, which focused on OVOCs only, no blanks were observed in their in-house zero gas."

Further, we added a plot into the supplemental (Fig. S1, supplemental p.7):

[Figure]

**Figure S1.** Peak areas for benzene, acetaldehyde and acetone observed at two different sample volumes: 1590ml and 390ml for benzene and 680ml and 1350ml for acetaldehyde and acetone. All peaks observed in the directly measured in-house zero
20   air (filled symbols) are independent of the sample volume. For acetaldehyde however, peaks observed in samples of in-house

zero gas which was flushed through the platinum catalyst at an early stage of usage (empty circles) are strongly affected by the sample volume.

5      Comment 4:

*Subchapters 3.1and 3.2 should be moved to a supplement. The typical reader of such a paper will not be interested in the details of peak evaluation and DL determination and knowledge of these details is not necessary to understand the results presented here.*

Response:

10     We agree and moved chapter 3 to the supplement.

Comment 5:

*Table 1 should be removed. Its content is only very indirectly connected to the subject of the paper and the information*

*presented has already been discussed in numerous publications and textbooks.*

15     Response:

We moved Table 1 into the supplement together with Section 3. Even though it sums up textbook information we think it valuable to keep it in the supplement with Section 3.

Comment 6:

20     *Table 2 should be moved to a supplement. The detection limits (as far as they are relevant) are obvious from Table 3 (I assume the <... indicates a concentration below the lower detection limit, a footnote explaining this should be added). If concentrations are above the DL the DL has little relevance for the findings presented here.*

Response:

We moved Table 2 to the supplement. In Table 3 (is now Table 1 after this revision) the "<…" indeed are concentrations
25     below the detection limit. This is already explained in the header of the table.

Comment 7:

*Table 3 should be separated into 3 tables (NMHC, Terpenes, OVOCS) which will avoid the many "empty" boxes. The "saved space" should be used to present the residual VOC concentrations for the use of "in-house zero gas" as feed for the*
30     *gas purifier as well as the VOC level in the "in-house" zero air without gas purifier. In my opinion, this information is of high interest for potential readers. What are the residual levels of VOC when using a "standard" combination of clean air*

*supply and a given gas purifier. A detail for Table 3 (and some other places in the paper), the number of significant digits presented should be consistent with the accuracy of the given data.*

Response:

We separated Table 3 like suggested and inserted the information about VOC concentrations in the in-house zero gas (step1 of the experiments) and VOC impurities released by the purifiers (step 2 of the experiments). We did not change the digits of the result data. As detection limits are in low pmol/mol range for the measured compounds these are the results we obtained from the raw data.

Comment 8:

*The finding that catalysts have to be "cleaned" by running for some time is not new, this part should be moved to a supplement. A useful information (if available) would be the time constant (if available) at which the different contaminants are removed from the catalysts, which would be relevant for justifying the two hour conditioning time used here.*

Response:

We removed this information from the conclusion part but left it in the experimental section. With the low time resolution of the gas chromatography technique used for the experiments of this paper we are unfortunately not able to provide the requested time constant. Typically in two hours we gain two gas chromatography measurements taking into account the time needed for sample enrichment and analysis with separation of the different VOCs on capillary columns. So, we just can give the information that zero gas was sufficiently clean (below detection limit) after two hours of flushing and heating the catalysts in this case. To specify the time constant a high time resolution instrument like a PTR-MS would be of an advantage.

Comment 9:

*Figure 6 and the detailed discussion of the results of Figure 6 should be moved to the supplement. Breakthrough as well as memory effects and dependence of the efficiency of adsorbents on humidity are nothing new. Moreover, it seems from Figure 7 that (even when averaging) giving a value for efficiency for the adsorbent is arbitrary since the result will (least for some of the < C5) depend on the duration of exposure to a feed with a given VOC level as well as the history of exposure to feeds with different VOC levels. The low efficiency for most <C5 HC and the high variability of efficiency for most C4 HC is evident from Table 3.*

Response:

We agree and moved Figures 6 and 7, and parts of the discussion to the supplement.

Comment 10:

*All chromatograms should be moved to a supplement. From the chromatograms I could not gain any important insight which is not already evident from Table 3.*

We moved the chromatograms from Fig. 2-5 to the supplement.

5   Comment 11:

*In the supplement the authors should provide linear regression information for the results obtained for step [1] in chapter 2.2 (sampling of different volumes) for compounds where the peak areas are not below the DL. This will allow readers interested in details to distinguish between "system blanks" and signals depending on sampled volume.*

Response:

10   We added a respective plot for benzene, acetaldehyde and acetone to the supplemental (p.7)

[Figure]

**Figure S1.** Peak areas for benzene, acetaldehyde and acetone observed at two different sample volumes: 1590ml and 390ml for benzene and 680ml and 1350ml for acetaldehyde and acetone. All peaks observed in the directly measured in-house zero air (filled symbols) are independent of the sample volume. For acetaldehyde however, peaks observed in samples of in-house

15   zero gas which was flushed through the platinum catalyst at an early stage of usage (empty circles) are strongly affected by the sample volume.

Comment 12:

*Conclusions: The first paragraph is mostly a summary of the introduction. It also contains statements that cannot be derived from the results presented her (e.g. the importance of monitoring blanks). This paragraph should be removed.*

Response:

We have modified the conclusion; parts of the first two paragraphs were removed or moved to the last paragraph (underlined below):

p.10, line 1ff:

"Two tested catalysts in this study were able to remove a large range of different VOCs. High mole fractions up to 50 nmol/mol were purified and residual concentrations were below the detection limits of the systems going down to less than 1 pmol/mol for NMHCs.

The tested adsorption cartridge was not suitable to remove light NMHCs ($C_2$ to $C_4$). There was a breakthrough behaviour of these compounds which was not constant. Also, VOC memory effects were observed. To characterise these effects repetition of measurements (> 5) would be of an advantage. However, it removed heavier VOCs, OVOCs and monoterpenes. An advantage of the adsorption cartridge is the lack of electricity. It could be a good alternative for applications where the breakthrough of light VOCs is of no relevance. A big disadvantage is the high influence of humidity on the lifetime of this kind of purifier. The tested model in this study was only adequate for use with very dry air up to maximum 1 µmol/mol water content. With this awareness it is highly recommended to enquire the maximum applicable water content of the used gas from the manufacturer of a purifier.

Finally, zero gas is often produced by compression of ambient air which constitutes a complex matrix with residual humidity. The cleaning process to receive high purity zero gases is a challenge to any purifying system. It is highly important to explicitly examine a gas purifier for its intended application. Tests should be done at the given conditions, e.g. the same flow rates and the same gas matrix with special focus on given target component concentrations and humidity. For the tests, measurement systems with adequate detection limits are essential. Potential internal blanks have to be detected and well characterised. Their long-term behaviour has to be controlled, especially for the enduring use in air quality monitoring stations".

Comment 1:

*In summary, the results presented are suitable to be published in AMT but as a Technical Note after streamlining the results presented together with an Electronic Supplement. The experiments conducted are tests with already available techniques for purifying air and thus no clear innovative approach is presented which may merit a publication as a full paper. The "introduction" should be condensed with finally leading to the research questions. The author team should justify their selection of the used purifying techniques.*

Response:

We now provide a supplement and the introduction was condensed. We removed the part describing several methods for generating hydrocarbon free air. Instead we just listed the different principles with some references for the interested reader.

p.2 lines 8ff:"Commonly used purification technologies in atmospheric monitoring include but are not limited to  gas purifiers based on inorganic media (e.g. Conte et al., 2008) or activated carbon (Van Osdell et al., 1996; Sircar et al., 1996), metal catalysts (Liotta, 2010; Heck et al., 2009) andphotocatalytical techniques (Debono et al, 2013; Huang et al., 2016) ."

 The research question we address is the application of suitable gas purifiers for ambient VOCs monitoring stations – a respective line was added into the last paragraph of the introduction, where the selection of purifying techniques was justified:

p.3 lines 1-3:

" In this study, three purifiers were selected to test their removal efficiency of a defined amount of VOCs to be applicable for ambient air monitoring stations. An adsorption cartridge with an inorganic media was selected for low-cost zero gas production without the need of electricity. In addition, the commonly used catalytic technique with an infinite lifespan has been tested for two types of catalyst."

Comment 2:

*In the "experimental" part all commercial suppliers of materials should be listed.*

Response:

The supplier of the adsorption cartridge did not agree to have the trademark published. We added a remark referring to this in p.3 line 6-7:

"…specified by the manufacturer (it was agreed not to publish the name and trademark)."

Comment 3:

*Details of the analytical systems used in the experiments should be reported in the supplement, likely in a Table.*

Response:

We provided the requested table with information on the used gas chromatography instruments in the supplement (Table S1).

Comment 4:

*The "data analytical details" chapter should be moved to the supplement as well as Table 1 and Table 2. A summary of the applied techniques is sufficient as part of the experimental/method section.*

Response:

We moved chapter 3 together with Table1 1and 2 to the supplement. In section 2, p.4, lines 1-2 we added the applied method how detection limits were derived:

"Detection limits for all systems were determined using IUPACs method based on the Neyman–Pearson theory of hypothesis testing (IUPAC, 1995, Section S2 in the supplement)."

Comment 5:

*The "results and discussion" part should focus on the performance of the purifying systems only. Additional information should be moved to the supplement. Figures 2 and 3 can be skipped. Figures 4, 5 and 6 may also be skipped or moved to the supplement, Figure 7 should be moved to the supplement.*

Response:

We condensed the chapter and moved Figures 2 to 7 to the supplement.

Comment 6:

*In the "conclusion" part of the paper please avoid duplication of the abstract and focus on aspects users of air purification systems should consider.*

Response:

We shortened the conclusion and focused on the main points of the paper:

p.10, line 1ff:

"Two tested catalysts in this study were able to remove a large range of different VOCs. High mole fractions up to 50 nmol/mol were purified and residual concentrations were below the detection limits of the systems going down to less than 1 pmol/mol for NMHCs.

The tested adsorption cartridge was not suitable to remove light NMHCs ($C_2$ to $C_4$). There was a breakthrough behaviour of these compounds which was not constant. Also, VOC memory effects were observed. To characterise these effects repetition of measurements ($> 5$) would be of an advantage. However, it removed heavier VOCs, OVOCs and monoterpenes. An advantage of the adsorption cartridge is the lack of electricity. It could be a good alternative for applications where the breakthrough of light VOCs is of no relevance. A big disadvantage is the high influence of humidity on the lifetime of this kind of purifier. The tested model in this study was only adequate for use with very dry air up to maximum 1 µmol/mol

water content. With this awareness it is highly recommended to enquire the maximum applicable water content of the used gas from the manufacturer of a purifier.

Finally, zero gas is often produced by compression of ambient air which constitutes a complex matrix with residual humidity. The cleaning process to receive high purity zero gases is a challenge to any purifying system. It is highly important
5 to explicitly examine a gas purifier for its intended application. Tests should be done at the given conditions, e.g. the same flow rates and the same gas matrix with special focus on given target component concentrations and humidity. For the tests, measurement systems with adequate detection limits are essential. Potential internal blanks have to be detected and well characterised. Their long-term behaviour has to be controlled, especially for the enduring use in air quality monitoring stations".

**Short comment by Marta Doval Miñarro**

*This paper presents the performance of several gas purifiers to produce zero gas for measuring VOCs. The availability of high purity and certified zero gases for atmospheric measurements is one of the issues that has to be tackled in order to*
15 *achieve further reductions in measurement uncertainty. The information presented in this paper is interesting and it moves towards this direction; however, I miss in the Introduction section a reference to previous work in this field. There are not many papers dealing with this topic so I find it necessary to cite the few of them in order to properly contextualise the manuscript of the authors. Some of the references I miss:*

*Miñarro et al. (2014) Zero gas reference standards. Analytical Methods 8, 15, 3014-3022.*

20 *Haerri H.P (2009) Trace gas analysis for the evaluation of zero air generators. Accreditation and Quality Assurance 14, 12, 647-654.*

Response:

We inserted the references suggested. Thanks for this important suggestion.

[revised manuscript text omitted]

Detection limits for all systems were determined using IUPACs method based on the Neyman–Pearson theory of hypothesis testing (IUPAC, 1995, Section S2 in the supplement).

**2.2 Experimental measurement setup and procedure**

30   For comparability a common procedure was applied by the three labs. Sample volumes used were dependent on the requirements of the different analysers. In general, sample volumes in between 400 ml to 3000 ml were applied. For all following steps a Rrepetition of five consecutive measurementsruns was 5 runsrecommended:

[1]     In step 1, tThe in-house zero gas was measured directly by the analysis systems. With this test, the labs to quantify its VOC impurities. Additionally all analysis systems were checked for internal blanks. eChecked for internal blanks

35         of their analysers system. Furthermore, with the sameand testschecked the in house zero gas used for the following steps of the experiment (2 to 4) could bewas checked for and in house VOC impurities and. for internal blanks of their analysers (VOC amounts) by measuring the in house zero gas (5 runs). The in house zero gas was measured directly by the analysis systems. Identification of internal blanks, i.e. system artefacts, and discrimination of them from zero gas impurities was done by measuring different sample volumes of the in-house zero gas. A proportional

40         relationship of the detector response with the sampled volume is expected for impurities in the in-house zero gas,

whereas for GC system internal blanks the detector response is expected to be independent of the sample volume.  The tested in-house zero gas was used for the following steps of the experiment (2 to 4).

[2]    In the next step the in-house zero gas from step 1 was supplied to one specific purifier to Checkquantify the VOC impurities originating from the  purifier itself.

[3]    In the third step the  efficiency of VOC removal of the tested purifier was checked by supplying a  VOC mixture  and measuring the outcome of residual VOCs.

[4]    In the last step the  incoming VOC concentration for step 3 was checked by supplying  the same preparation of VOC mixture directly to the analysis system (no purifying).

After step four a repetition of steps one and two was optional for the labs but is advisable to monitor the status of the set-up.

A unified flow rate of 1 slpm was applied being within the specification of each purifier model. The two catalysts were heated and flushed with zero gas for at least two hours before starting the experiments. This was needed to reduce VOC impurities originating from the catalysts being freshly installed. The experimental set-up is shown in Figure 1.

[Figure]

**Figure 1.** Experimental set-up for testing the purifier performance.

Test mixtures with different VOC mole fractions were produced by dynamic generation methods, e.g. dilution of high concentrated static VOC mixtures in cylinders (Figure 1) or diffusion methods (Demichelis, 2016). Following test mixtures were supplied: NMHCs at 1.2, 5 and 50 nmol/mol, monoterpenes at 1.2 nmol/mol, OVOCs from 10 to 70 nmol/mol and acetonitrile at 10 nmol/mol. For the in-house zero gas DWD used compressed and dried (water content ~ 1000 µmol/mol) ambient air purified by a palladium catalyst. VSL and INRIM used synthetic air cylinders (grade 6.0, water content < 0.5 µmol/mol, total hydrocarbons content < 0.05 µmol/mol).

**3 Results and discussion**

To ensure comparability between the participating groups the same measurement procedure described in Sect. 2.2 has been applied. All GC chromatograms were analysed visually. Peaks of VOCs in the chromatograms were integrated by GC software and mole fraction were subsequently determined for each single measurement and average mole fractions and standard deviations, respectively, were derived for each measurements series (Table 13).

Before assessing the purifier efficiency, in-house zero gas quality and internal blanks were determined by steps one of the measurement procedure (Sect. 2.2.). The results for VSL and DWD are shown in the first two columns in Table1. In the DWD in-house zero air all substances were below the detection limit, with exception for benzene (4pmol/mol), acetaldehyde (124pmol/mol) and acetone (52pmol/mol). The observed peaks were independent of the sample volume (see supplemental Fig. S1and S2), and showed the characteristics of an internal blank and are not regarded as an impurity of the DWD in-house zero gas. For VSL, blank values were observed at a level of 20-50 pmol/mol for several alkanes (Table 1). This is The results are consistent within the specification of the used synthetic air grade 6.0, which allowings up to 50 nmol/mol of hydrocarbons. This but also highleighgt ts the need for further purification of commercial cylinders in order to yield a hig quality zero airto assure low impurities levels for high quality zero air . With the INRIM system, which focussed on OVOCs only, no blanks were observed in their in-house zero gas. .

Subsequently, VOC release of the purifier itself was checked (step 2 in Sect 2.2). E.g. the platinum catalyst showed acetaldehyde impurities scaled with the sample volume (Fig. S1 and S31). By flushing the catalysts for two hours with zero air (1l/min), the relevant impurities were below the detection limits.

[revised manuscript text omitted]

**Residual monoterpenes [pmol/mol]:**

| Purifier: | In-house zero | Adsorption cartridge | | Palladium catalyst | | Platinum catalyst | |
|---|---|---|---|---|---|---|---|
| Testing lab: | DWD | DWD | | DWD | | DWD | |
| Supplied amount of m.terpenes [nmol/mol]: | 0 | 0 | 1.2 | 0 | 1.2 | 0 | 1.2 |
| alpha-pinene | < 4 | < 4 | < 4 | < 4 | < 4 | < 4 | < 4 |
| myrcene | < 3 | < 3 | < 3 | < 3 | < 3 | < 3 | < 3 |
| 3-carene | < 2 | < 2 | < 2 | < 2 | < 2 | < 2 | < 2 |
| cis-ocimene | < 2 | < 2 | < 2 | < 2 | < 2 | < 2 | < 2 |
| p-cymene | < 2 | < 2 | < 2 | < 2 | < 2 | < 2 | < 2 |
| limonene | < 2 | < 2 | < 2 | < 2 | < 2 | < 2 | < 2 |
| camphor | < 2 | < 2 | < 2 | < 2 | < 2 | < 2 | < 2 |
| 1,8-cineole | < 5 | < 5 | < 5 | < 5 | < 5 | < 5 | < 5 |

**Residual OVOCs and acetonitrile [pmol/mol]:**

| Purifier: | In-house zero gas (air) | | | Adsorption cartridge | | | | | | Palladium catalyst | | | | | | Platinum catalyst | |
|---|---|---|---|---|---|---|---|---|---|---|---|---|---|---|---|---|---|
| Testing lab: | DWD | VSL | INRIM | DWD | | VSL | | INRIM | | DWD | | VSL | | INRIM | | DWD | |
| Supplied amount of OVOCs [nmol/mol]: | 0 | 0 | 0 | 0 | 10 | 0 | 10 | 0 | 20 to 70 | 0 | 10 | 0 | 10 | 0 | 20 to 70 | 0 | 10 |
| methanol | < 77 | < 110 | < 3 | < 77 | < 77 | < 110 | < 110 | n.a. | 67 ± 11 | < 77 | < 77 | < 110 | < 110 | n.a. | 73 ± 10 | < 77 | < 77 |
| acetaldehyde | 124 ± 19 | < 110 | n.a. | < 84 | < 84 | < 110 | < 110 | n.a. | n.a. | < 84 | < 84 | < 110 | < 110 | n.a. | n.a. | < 84 | < 84 |
| ethanol | < 26 | < 120 | < 11 | < 26 | < 26 | < 120 | < 120 | n.a. | < 11 | < 26 | < 26 | < 120 | < 120 | n.a. | < 11 | < 26 | < 26 |
| acetone | 52 ± 15 | < 80 | < 11 | < 31 | < 31 | < 80 | < 80 | n.a. | 57 ± 10 | < 31 | < 31 | < 80 | < 80 | n.a. | 63 ± 9 | < 31 | < 31 |
| MEK | < 2 | < 180 | n.a. | < 2 | < 2 | < 180 | < 180 | n.a. | n.a. | < 2 | < 2 | < 180 | < 180 | n.a. | n.a. | < 2 | < 2 |
| methacrolein | n.a. | < 110 | n.a. | n.a. | n.a. | < 110 | < 110 | n.a. | n.a. | n.a. | n.a. | < 110 | < 110 | n.a. | n.a. | n.a. | n.a. |
| acetonitrile | < 6 | n.a. | n.a. | < 6 | < 6 | n.a. | n.a. | n.a. | n.a. | < 6 | < 6 | n.a. | n.a. | n.a. | n.a. | < 6 | < 6 |

**4 Conclusions**

[revised manuscript text omitted]